# Selective Encoding for Abstractive Sentence Summarization

## Abstract

We propose a selective encoding model to extend the sequence-to-sequence framework for abstractive sentence summarization. It consists of a sentence encoder, a selective gate network, and an attention equipped decoder. The sentence encoder and decoder are built with recurrent neural networks. The selective gate network constructs a second level sentence representation by controlling the information flow from encoder to decoder. The second level representation is tailored for sentence summarization task, which leads to better performance. We evaluate our model on the English Gigaword, DUC 2004 and MSR abstractive sentence summarization datasets. The experimental results show that the proposed selective encoding model outperforms the state-of-the-art baseline models.

## 1 Introduction

Sentence summarization aims to shorten a given sentence and produce a brief summary of it. This is different from document level summarization task since it is hard to apply existing techniques in extractive methods, such as extracting sentence level features and ranking sentences. Early works propose using rule-based methods (Zajic et al., 2007), syntactic tree pruning methods (Knight and Marcu, 2002), statistical machine translation techniques (Banko et al., 2000) and so on for this task. We focus on abstractive sentence summarization task in this paper.

Recently, neural network models have been applied in this task. Rush et al. (2015) use auto-constructed sentence-headline pairs to train a neural network summarization model. They use a Convolutional Neural Network (CNN) encoder and feed-forward neural network language model decoder for this task. Chopra et al. (2016) extend their work by replacing the decoder with Recurrent Neural Network (RNN). Nallapati et al. (2016) follow this line and change the encoder to RNN to make it a full RNN based sequence-to-sequence model (Sutskever et al., 2014).

> the **sri lankan** government on wednesday announced the **closure** of **government schools** with **immediate effect** as a **military campaign** against **tamil separatists escalated** in the north of the country .

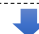

> sri lanka closes schools as war escalates

Figure 1: An abstractive sentence summarization system may produce the output summary by distilling the salient information from the highlight to generate a fluent sentence. We model the distilling process with selective encoding.

All the above works fall into the encoding-decoding paradigm, which first encodes the input sentence to an abstract representation and then decodes the intended output sentence based on the encoded information. As an extension of the encoding-decoding framework, attention-based approach (Bahdanau et al., 2015) has been broadly used: the encoder produces a list of vectors for all tokens in the input, and the decoder uses an attention mechanism to dynamically extract encoded information and align with the output tokens. This approach achieves huge success in tasks like neural machine translation, where alignment between all parts of the input and output are required. However, in abstractive sentence summarization, there is no explicit alignment relationship between input sentence and the summary except for the extracted common words. The

challenge here is not to infer the alignment, but to select the highlights while filtering out secondary information in the input. A desired work-flow for abstractive sentence summarization is encoding, selection, and decoding. After selecting the important information from an encoded sentence, the decoder produces the output summary using the selected information. For example, in Figure 1, given the input sentence, the summarization system first selects the important information, and then rephrases or paraphrases to produce a well-organized summary. Although this is implicitly modeled in the encoding-decoding framework, we argue that abstractive sentence summarization shall benefit from explicitly modeling this selection process.

In this paper we propose **S**elective **E**ncoding for **A**bstractive **S**entence **S**ummarization (**SEASS**). We treat the sentence summarization as a three-phase task: encoding, selection, and decoding. It consists of a sentence encoder, a selective gate network, and a summary decoder. First, the sentence encoder reads the input words through an RNN unit to construct the first level sentence representation. Then the selective gate network selects the encoded information to construct the second level sentence representation. The selective mechanism controls the information flow from encoder to decoder by applying a gate network according to the sentence information, which helps improve encoding effectiveness and release the burden of the decoder. Finally, the attention-equipped decoder generates the summary using the second level sentence representation. We conduct experiments on English Gigaword, DUC 2004 and Microsoft Research Abstractive Text Compression test sets. Our SEASS model achieves 17.54 ROUGE-2 F1, 9.56 ROUGE-2 recall and 10.63 ROUGE-2 F1 on these test sets respectively, which improves performance compared to the state-of-the-art methods.

## 2 Problem Formulation

For sentence summarization, given an input sentence $x = (x_1, x_2, \ldots, x_n)$, where $n$ is the sentence length, $x_i \in \mathcal{V}_s$ and $\mathcal{V}_s$ is the source vocabulary, the system summarizes $x$ by producing $y = (y_1, y_2, \ldots, y_l)$, where $l \leq n$ is the summary length , $y_i \in \mathcal{V}_t$ and $\mathcal{V}_t$ is the target vocabulary.

If $|y| \subseteq |x|$, which means all words in summary $y$ must appear in given input, we denote this as *extractive* sentence summarization. If $|y| \nsubseteq |x|$,

which means not all words in summary come from input sentence, we denote this as *abstractive* sentence summarization. Table 1 provides an example. We focus on *abstracive* sentence summarization task in this paper.

| Input: | South Korean President Kim Young-Sam left here Wednesday on a week - long state visit to Russia and Uzbekistan for talks on North Korea 's nuclear confrontation and ways to strengthen bilateral ties . |
|---|---|
| Output: | Kim leaves for Russia for talks on NKorea nuclear standoff |

Table 1: An abstractive sentence summarization example.

## 3 Model

As shown in Figure 2, our model consists of a sentence encoder using the Gated Recurrent Unit (GRU) (Cho et al., 2014), a selective gate network and an attention-equipped GRU decoder. First, the bidirectional GRU encoder reads the input words $x = (x_1, x_2, \ldots, x_n)$ and builds its representation $(h_1, h_2, \ldots, h_n)$. Then the selective gate selects and filters the word representations according to the sentence meaning representation to produce a tailored sentence word representation for abstractive sentence summarization task. Lastly, the GRU decoder produces the output summary with attention to the tailored representation. In the following sections, we introduce the sentence encoder, the selective mechanism, and the summary decoder respectively.

### 3.1 Sentence Encoder

The role of the sentence encoder is to read the input sentence and construct the basic sentence representation. Here we employ a bidirectional GRU (BiGRU) as the recurrent unit, where GRU is defined as:

$$z_i = \sigma(\mathbf{W}_z[x_i, h_{i-1}]) \qquad (1)$$

$$r_i = \sigma(\mathbf{W}_r[x_i, h_{i-1}]) \qquad (2)$$

$$\widetilde{h}_i = \tanh(\mathbf{W}_h[x_i, r_i \odot h_{i-1}]) \qquad (3)$$

$$h_i = (1 - z_i) \odot h_{i-1} + z_i \odot \widetilde{h}_i \qquad (4)$$

The BiGRU consists of a forward GRU and a backward GRU. The forward GRU reads the input sentence word embeddings from left to right and gets a sequence of hidden states, $(\vec{h}_1, \vec{h}_2, \ldots, \vec{h}_n)$. The backward GRU reads the input sentence

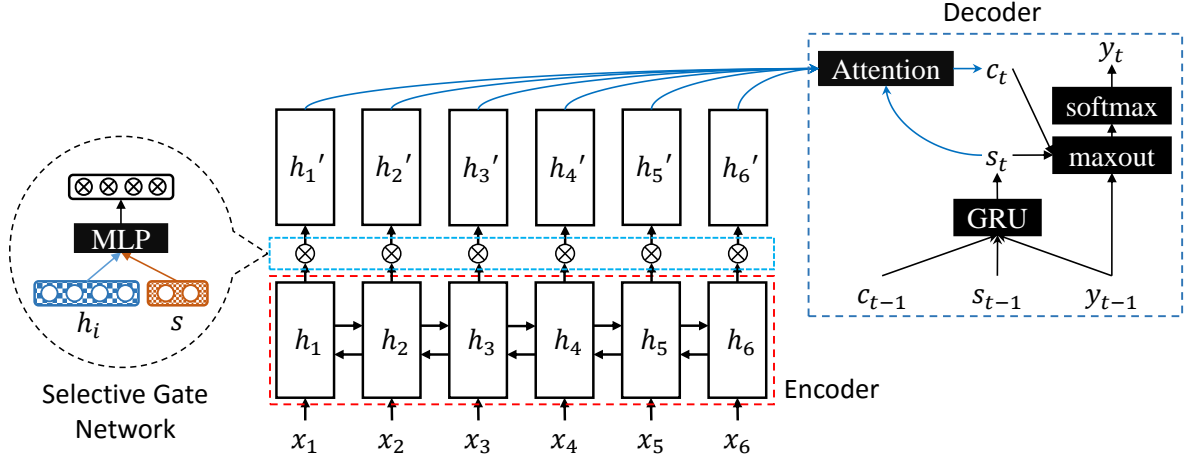

Figure 2: Overview of the **S**elective **E**ncoding for **A**bstractive **S**entence **S**ummarization (**SEASS**).

embeddings reversely, from right to left, and results in another sequence of hidden states, $(\overleftarrow{h}_1, \overleftarrow{h}_2, \ldots, \overleftarrow{h}_n)$:

$$\vec{h}_i = \text{GRU}(x_i, \vec{h}_{i-1}) \tag{5}$$
$$\overleftarrow{h}_i = \text{GRU}(x_i, \overleftarrow{h}_{i+1}) \tag{6}$$

The initial states of the BiGRU are set to zero vectors, i.e., $\vec{h}_1 = 0$ and $\overleftarrow{h}_n = 0$. After reading the sentence, the forward and backward hidden states are concatenated, i.e., $h_i = [\vec{h}_i; \overleftarrow{h}_i]$, to get the basic sentence representation.

### 3.2 Selective Mechanism

In the sequence-to-sequence machine translation (MT) model, the encoder and decoder are responsible for encoding input sentence information and decoding the sentence representation to generate an output sentence. Some previous works apply this framework to summarization generation tasks. However, abstractive sentence summarization is different from MT in two ways. First, there is no explicit alignment relationship between the input sentence and the output summary except for the common words. Second, summarization task needs to keep the highlights and remove the unnecessary information, while MT needs to keep all information literally.

Herein, we propose a selective mechanism to model the selection process for abstractive sentence summarization. The selective mechanism extends the sequence-to-sequence model by constructing a tailored representation for abstractive sentence summarization task. Concretely, the selective gate network in our model takes two vector

inputs, the sentence word vector $h_i$ and the sentence representation vector $s$. The sentence word vector $h_i$ is the output of the BiGRU encoder and represents the meaning and context information of word $x_i$. The sentence vector $s$ is used to represent the meaning of the sentence. For each word $x_i$, the selective gate network generates a gate vector $sGate_i$ using $h_i$ and $s$, then the tailored representation is constructed, i.e., $h_i'$.

In detail, we concatenate the last forward hidden state $\vec{h}_n$ and backward hidden state $\overleftarrow{h}_1$ as the sentence representation $s$:

$$s = \begin{bmatrix} \overleftarrow{h}_1 \\ \vec{h}_n \end{bmatrix} \tag{7}$$

For each time step $i$, the select gate takes the sentence representation $s$ and BiGRU hidden $h_i$ as inputs to compute the select gate $sGate_i$:

$$sGate_i = \sigma(\mathbf{W}_s h_i + \mathbf{U}_s s + b) \tag{8}$$
$$h_i' = h_i \odot sGate_i \tag{9}$$

where $\sigma$ is sigmoid activation function and $\odot$ is element-wise multiplication. After the selective gate network, we obtain another sequence of vectors $(h_1', h_2', \ldots, h_n')$. This new sequence is then used as the input sentence representation for the decoder to generate the summary.

### 3.3 Summary Decoder

On top of the sentence encoder and the selective gate network, we use GRU with attention as the decoder to produce the output summary.

At each decoding time step $t$, the GRU reads the previous word embedding $w_{t-1}$ and previous context vector $c_{t-1}$ as inputs to compute the new hidden state $s_t$. To initialize the GRU hidden state, we use a linear layer with the last backward encoder hidden state $\overleftarrow{h}_1$ as input:

$$s_t = \text{GRU}(w_{t-1}, c_{t-1}, s_{t-1}) \quad (10)$$

$$s_0 = \tanh(\mathbf{W}_d \overleftarrow{h}_1 + b) \quad (11)$$

The context vector $c_t$ for current time step $t$ is computed through the concatenate attention mechanism (Luong et al., 2015), which matches the current decoder state $s_t$ with each encoder hidden state $h'_i$ to get an importance score. The importance scores are then normalized to get the current context vector by weighted sum:

$$e_{t,i} = v_a^\top \tanh(\mathbf{W}_a s_{t-1} + \mathbf{U}_a h'_i) \quad (12)$$

$$\alpha_{t,i} = \frac{\exp(e_{t,i})}{\sum_{i=1}^n \exp(e_{t,i})} \quad (13)$$

$$c_t = \sum_{i=1}^n \alpha_{t,i} h'_i \quad (14)$$

We then combine the previous word embedding $w_{t-1}$, the current context vector $c_t$, and the decoder state $s_t$ to construct the readout state $r_t$. The readout state is then passed through a maxout hidden layer (Goodfellow et al., 2013) to predict the next word with a softmax layer over the decoder vocabulary.

$$r_t = \mathbf{W}_r w_{t-1} + \mathbf{U}_r c_t + \mathbf{V}_r s_t \quad (15)$$

$$m_t = [\max\{r_{t,2j-1}, r_{t,2j}\}]_{j=1,\dots,d}^\top \quad (16)$$

$$p(y_t | y_1, \dots, y_{t-1}) = \text{softmax}(\mathbf{W}_o m_t) \quad (17)$$

where $r_t$ is a $2d$-dimensional vector.

### 3.4 Objective Function

Our goal is to maximize the output summary probability given the input sentence. Therefore, we optimize the negative log-likelihood loss function:

$$J(\theta) = -\frac{1}{|\mathcal{D}|} \sum_{(x,y) \in \mathcal{D}} \log p(y|x) \quad (18)$$

where $\mathcal{D}$ denotes a set of parallel sentence-summary pairs and $\theta$ is the model parameter. We use Stochastic Gradient Descent (SGD) with mini-batch to learn the model parameter $\theta$.

## 4 Experiments

In this section we introduce the dataset we use, the evaluation metric, the implementation details, the baselines we compare to, and the performance of our system.

### 4.1 Dataset

**Training Set** For our training set, we use a parallel corpus which is constructed from the Annotated English Gigaword dataset (Napoles et al., 2012) as mentioned in Rush et al. (2015). The parallel corpus is produced by pairing the first sentence and the headline in the news article with some heuristic rules. We use the script[1] released by Rush et al. (2015) to pre-process and extract the training and development datasets. The script performs various basic text normalization, including PTB tokenization, lower-casing, replacing all digit characters with #, and replacing word types seen less than 5 times with $\langle unk \rangle$. The extracted corpus contains about 3.8M sentence-summary pairs for the training set and 189K examples for the development set.

For our test set, we use the English Gigaword, DUC 2004, and Microsoft Research Abstractive Text Compression test sets.

**English Gigaword Test Set** We randomly sample 8000 pairs from the extracted development set as our development set since it is relatively large. For the test set, we use the same randomly held-out test set of 2000 sentence-summary pairs as Rush et al. (2015).[2]

We also find that except for the empty titles, this test set has some invalid lines like the input sentence containing only one word. Therefore, we further sample 2000 pairs as our internal test set and release it for future works.

**DUC 2004 Test Set** We employ DUC 2004 data for tasks 1 & 2 (Over et al., 2007) in our experiments as one of the test sets since it is too small to train a neural network model on. The dataset pairs each document with 4 different human-written reference summaries which are capped at 75 bytes. It has 500 input sentences with each sentence paired with 4 summaries.

---

[1] https://github.com/facebook/NAMAS

[2] Thanks to Rush et al. (2015), we acquired the test set they used. Following Chopra et al. (2016), we remove pairs with empty titles resulting in slightly different accuracy compared to Rush et al. (2015) for their systems. The cleaned test set contains 1951 sentence-summary pairs.

**MSR-ATC Test Set**  Toutanova et al. (2016) release a new dataset for sentence summarization task by crowdsourcing. This dataset contains approximately 6,000 source text sentences with multiple manually-created summaries (about 26,000 sentence-summary pairs in total). Toutanova et al. (2016) provide a standard split of the data into training, development, and test sets, with 4,936, 448 and 785 input sentences respectively. Since the training set is too small, we only use the test set as one of our test sets. We denote this dataset as MSR-ATC (Microsoft Research Abstractive Text Compression) test set in the following.

Table 2 summarizes the statistic information of the three datasets we used.

| Data Set | Giga | DUC$^{\dagger}$ | MSR$^{\dagger}$ |
|---|---|---|---|
| #(sent) | 3.99M | 500 | 785 |
| #(sentWord) | 125M | 17.8K | 29K |
| #(summWord) | 33M | 20.9K | 85.9K |
| #(ref) | 1 | 4 | 3-5 |
| avgInputLen | 31.35 | 35.56 | 36.97 |
| avgSummLen | 8.23 | 10.43 | 25.5 |

Table 2: Data statistics for the datasets. †DUC 2004 and MSR-ATC are for test purpose only.

### 4.2 Evaluation Metric

We employ ROUGE (Lin, 2004) as our evaluation metric. ROUGE measures the quality of summary by computing overlapping lexical units, such as unigram, bigram, trigram, and longest common subsequence (LCS). It becomes the standard evaluation metric for DUC shared tasks and popular for summarization evaluation. Following previous work, we use ROUGE-1 (unigram), ROUGE-2 (bigram) and ROUGE-L (LCS) as the evaluation metrics in the reported experimental results.

### 4.3 Implementation Details

**Model Parameters**  The input and output vocabularies are collected from the training data, which have 119,504 and 68,883 word types respectively. We set the word embedding size to 300 and all GRU hidden state sizes to 512. We use dropout (Srivastava et al., 2014) with probability $p = 0.5$.

**Model Training**  We initialize model parameters randomly using a Gaussian distribution with Xavier scheme (Glorot and Bengio, 2010). We use a combination of Adam (Kingma and Ba, 2015) and simple SGD as our the optimizing algorithms. The training is separated into two phases, the first phase is optimizing the loss function with Adam and the second is with simple SGD. For the Adam optimizer, we set the learning rate $\alpha = 0.001$, two momentum parameters $\beta_1 = 0.9$ and $\beta_2 = 0.999$ respectively, and $\epsilon = 10^{-8}$. We use Adam optimizer until the training perplexity decrease to threshold 50. Then we switch to a simple SGD optimizer with initial learning rate $\alpha = 0.5$ and halve it for every 300K batches. We also apply gradient clipping (Pascanu et al., 2013) with range $[-5, 5]$ for both Adam and SGD phases. To both speed up the training and converge quickly, we use mini-batch size 64 by grid search.

**Beam Search**  We use beam search to generate multiple summary candidates to get better results. To avoid favoring shorter outputs, we average the ranking score along the beam path by dividing it by the number of generated words. To both decode fast and get better results, we set the beam size to 12 in our experiments.

### 4.4 Baseline

We compare SEASS model with the following state-of-the-art baselines:

**ABS**  Rush et al. (2015) use a CNN encoder and NNLM decoder to do the sentence summarization task.

**ABS+**  Based on ABS model, Rush et al. (2015) further tune their model using DUC 2003 dataset, which leads to improvements on DUC 2004 test set.

**CAs2s**  As an extension of the ABS model, Chopra et al. (2016) use a convolutional attention-based encoder and RNN decoder, which outperforms the ABS model.

**Feats2s**  Nallapati et al. (2016) use a full RNN sequence-to-sequence encoder-decoder model and add some features to enhance the encoder, such as POS tag, NER, and so on.

**Luong-NMT**  Neural machine translation model of Luong et al. (2015) with two-layer LSTMs for the encoder-decoder with 500 hidden units in each layer implemented in (Chopra et al., 2016).

**s2s+att**  We also implement a sequence-to-sequence model with attention as our baseline and denote it as "s2s+att".

## 4.5 Results

We report ROUGE F1, ROUGE recall and ROUGE F1 for English Gigaword, DUC 2004 and MSR-ATC test sets respectively. We use the official ROUGE script (version 1.5.5) [3] to evaluate the summarization quality in our experiments. For English Gigaword[4] and MSR-ATC[5] test sets, the outputs have different lengths so we evaluate the system with F1 metric. As for the DUC 2004 test set[6], the task requires the system to produce a fixed length summary (75 bytes), therefore we employ ROUGE recall as the evaluation metric. To satisfy the length requirement, we decode the output summary to a roughly expected length following Rush et al. (2015).

**English Gigaword** We acquire the test set from Rush et al. (2015) so we can make fair comparisons to the baselines.

| Models | RG-1 | RG-2 | RG-L |
|---|---|---|---|
| ABS (beam)[‡] | 29.55[-] | 11.32[-] | 26.42[-] |
| ABS+ (beam)[‡] | 29.76[-] | 11.88[-] | 26.96[-] |
| Feats2s (beam)[‡] | 32.67[-] | 15.59[-] | 30.64[-] |
| CAs2s (greedy)[‡] | 33.10[-] | 14.45[-] | 30.25[-] |
| CAs2s (beam)[‡] | 33.78[-] | 15.97[-] | 31.15[-] |
| Luong-NMT (beam)[‡] | 33.10[-] | 14.45[-] | 30.71[-] |
| s2s+att (greedy) | 33.18[-] | 14.79[-] | 30.80[-] |
| s2s+att (beam) | 34.04[-] | 15.95[-] | 31.68[-] |
| SEASS (greedy) | 35.48 | 16.50 | 32.93 |
| SEASS (beam) | **36.15** | **17.54** | **33.63** |

Table 3: Full length ROUGE F1 evaluation results on the English Gigaword test set used by Rush et al. (2015). RG in the Table denotes ROUGE. Results with [‡] mark are taken from the corresponding papers. The supper script [-] indicates that our SEASS model with beam search performs significantly better than it as given by the 95% confidence interval in the official ROUGE script.

In Table 3, we report the ROUGE F1 score of our model and the baseline methods. Our SEASS model with beam search outperforms all baseline models by a large margin. Even for greedy search, our model still performs better than other methods

[3]http://www.berouge.com/
[4]The ROUGE evaluation option is the same as Rush et al. (2015), -m -n 2 -w 1.2
[5]The ROUGE evaluation option is, -m -n 2 -w 1.2
[6]The ROUGE evaluation option is, -m -b 75 -n 2 -w 1.2

| Models | RG-1 | RG-2 | RG-L |
|---|---|---|---|
| ABS (beam) | 37.41[-] | 15.87[-] | 34.70[-] |
| s2s+att (greedy) | 42.41[-] | 20.76[-] | 39.84[-] |
| s2s+att (beam) | 43.76[-] | 22.28[-] | 41.14[-] |
| SEASS (greedy) | 45.27 | 22.88 | 42.20 |
| SEASS (beam) | **46.86** | **24.58** | **43.53** |

Table 4: Full length ROUGE F1 evaluation on our internal English Gigaword test data. The supper script [-] indicates that our SEASS model performs significantly better than it as given by the 95% confidence interval in the official ROUGE script.

which used beam search. For the popular ROUGE-2 metric, our SEASS model achieves 17.54 F1 score and performs better than the previous works. Compared to the ABS model, our model has a 6.22 ROUGE-2 F1 relative gain. Compared to the highest CAs2s baseline, our model achieves 1.75 ROUGE-2 F1 improvement and passes the significant test according to the official ROUGE script.

Table 4 summarizes our results on our internal test set using ROUGE F1 evaluation metrics. The performance on our internal test set is comparable to our development set, which achieves 24.58 ROUGE-2 F1 and outperforms the baselines.

| Models | RG-1 | RG-2 | RG-L |
|---|---|---|---|
| ABS (beam)[‡] | 26.55[-] | 7.06[-] | 22.05[-] |
| ABS+ (beam)[‡] | 28.18[-] | 8.49[-] | 23.81[-] |
| Feats2s (beam)[‡] | 28.35[-] | 9.46 | 24.59[-] |
| CAs2s (greedy)[‡] | 29.13 | 7.62[-] | 23.92[-] |
| CAs2s (beam)[‡] | 28.97 | 8.26[-] | 24.06[-] |
| Luong-NMT (beam)[‡] | 28.55 | 8.79[-] | 24.43[-] |
| s2s+att (greedy) | 27.03[-] | 7.89[-] | 23.80[-] |
| s2s+att (beam) | 28.13 | 9.25 | 24.76 |
| SEASS (greedy) | 28.68 | 8.55 | 25.04 |
| SEASS (beam) | **29.21** | **9.56** | **25.51** |

Table 5: ROUGE recall evaluation results on DUC 2004 test set. All these models are tested using beam search. Results with [‡] mark are taken from the corresponding papers. The supper script [-] indicates that our SEASS model performs significantly better than it as given by the 95% confidence interval in the official ROUGE script.

**DUC 2004** We evaluate our model using the ROUGE recall score since the reference summaries

of the DUC 2004 test set are capped at 75 bytes. Therefore, we decode the summary to a fixed length 18 to ensure that the generated summary satisfies the minimum length requirement. As summarized in Table 5, our SEASS outperforms all the baseline methods and achieves 29.21, 9.56 and 25.51 for ROUGE 1, 2 and L recall. Compared to the ABS+ model which is tuned using DUC 2003 data, our model performs significantly better by 1.07 ROUGE-2 recall score and is trained only with English Gigaword sentence-summary data without being tuned using DUC data.

**MSR-ATC** We report the full length ROUGE F1 score on the MSR-ATC test set in Table 6. To the best of our knowledge, this is the first work that reports ROUGE metric scores on the MSR-ATC dataset. Note that we only compare our model with ABS since the others are not publicly available. Our SEASS achieves 10.63 ROUGE-2 F1 and outperforms the s2s+att baseline by 1.02 points.

| Models | RG-1 | RG-2 | RG-L |
|---|---|---|---|
| ABS (beam) | 20.27⁻ | 5.26⁻ | 17.10⁻ |
| s2s+att (greedy) | 15.15⁻ | 4.48⁻ | 13.62⁻ |
| s2s+att (beam) | 22.65⁻ | 9.61⁻ | 21.39⁻ |
| SEASS (greedy) | 19.77 | 6.44 | 17.36 |
| SEASS (beam) | **25.75** | **10.63** | **22.90** |

Table 6: Full length ROUGE F1 evaluation on MSR-ATC test set. Beam search are used in both the baselines and our method. The supper script ⁻ indicates that our SEASS model performs significantly better than it as given by the 95% confidence interval in the official ROUGE script.

## 5 Discussion

In this section, we first compare the performance of SEASS with the s2s+att baseline model to illustrate that the proposed method succeeds in selecting information and building tailored representation for abstractive sentence summarization. We then analyze selective encoding by visualizing the heat map.

**Effectiveness of Selective Encoding** We further test the SEASS model with different sentence lengths on English Gigaword test sets, which are merged from the Rush et al. (2015) test set and our internal test set. The length of sentences in the test sets ranges from 10 to 80. We group the sentences

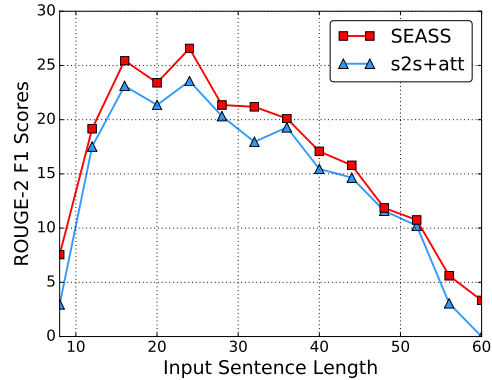

Figure 3: ROUGE-2 F1 score on different groups of input sentences in terms of their length for s2s+att baseline and our SEASS model on English Gigaword test sets.

with an interval of 4 and get 18 different groups and we draw the first 14 groups. We find that the performance curve of our SEASS model always appears to be on the top of that of s2s+att with a certain margin. For the groups of 16, 20, 24, 32, 56 and 60, the SEASS model obtains big improvements compared to the s2s+att model. Overall, these improvements on all groups indicate that the selective encoding method benefits the abstractive sentence summarization task.

**Saliency Heat Map of Selective Gate** Since the output of the selective gate network is a high dimension vector, it is hard to visualize all the gate values. We use the method in Li et al. (2016) to visualize the contribution of the selective gate to the final output, which can be approximated by the first derivative. Given sentence words $x$ with associated output summary $y$, the trained model associates the pair $(x, y)$ with a score $S_y(x)$. The goal is to decide which gate $g$ associated with $x$ makes the most significant contribution to $S_y(x)$. We approximate the $S_y(g)$ by computing the first-order Taylor expansion

$$S_y(g) \approx \frac{\partial(S_y)}{\partial g} g + b \qquad (19)$$

We then draw the Euclidean norm of the first derivative of the output $y$ with respect to the selective gate $g$ associated with each input words.

Figure 4 shows an example of the first derivative heat map, in which most of the important words are selected by the selective gate such as "europe", "slammed", "unacceptable", "conditions",

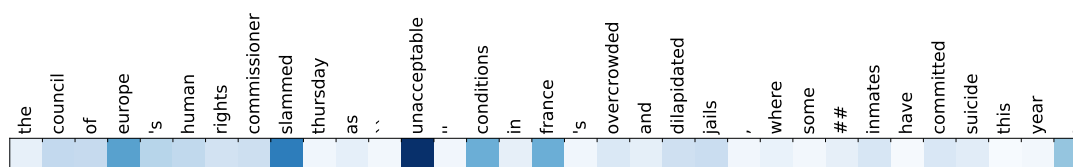

Figure 4: First derivative heat map of the output with respect to the selective gate. The important words are selected in the input sentence, such as "europe", "slammed" and "unacceptable". The output summary of our system is "council of europe slams french prison conditions" and the true summary is "council of europe again slams french prison conditions".

and "france". We can observe that the selective gate determines the importance of each word before decoder, which releases the burden of it by providing tailored sentence encoding.

## 6 Related Work

Abstractive sentence summarization, also known as sentence compression and similar to headline generation, is used to help compress or fuse the selected sentences in extractive document summarization systems since they may inadvertently include unnecessary information. There are some previous methods to solve this task, such as the linguistic rule-based method (Dorr et al., 2003). As for the statistical machine learning based methods, Banko et al. (2000) apply statistical machine translation techniques by modeling headline generation as a translation task and use 8000 article-headline pairs to train the system.

Rush et al. (2015) propose leveraging news data in Annotated English Gigaword (Napoles et al., 2012) corpus to construct large scale parallel data for sentence summarization task. They propose an ABS model, which consists of an attentive Convolutional Neural Network encoder and an neural network language model (Bengio et al., 2003) decoder. Chopra et al. (2016) extend this work, which keeps the CNN encoder but replaces the decoder with recurrent neural networks. Nallapati et al. (2016) further change the encoder to an RNN encoder, which leads to a full RNN sequence-to-sequence model. Besides, they enrich the encoder with lexical and statistic features which play important roles in traditional feature based summarization systems, such as NER and POS tags, to improve performance. Experiments on the Gigaword and DUC 2004 test sets show that the above models achieve state-of-the-art results.

Gu et al. (2016) and Gulcehre et al. (2016) come up similar ideas that summarization task can benefit from copying words from input sentences. Gu et al. (2016) propose CopyNet to model the copying action in response generation, which also applies for summarization task. Gulcehre et al. (2016) propose a switch gate to control whether to copy from source or generate from decoder vocabulary. Zeng et al. (2016) also propose using copy mechanism and add an extra gate on the GRU/LSTM gate for this task. Cheng and Lapata (2016) use an RNN based encoder-decoder for extractive summarization of documents

Yu et al. (2016) propose a segment to segment neural transduction model for sequence-to-sequence framework. The model introduces a latent segmentation which determines correspondences between tokens of the input sequence and the output sequence. Shen et al. (2016) propose applying Minimum Risk Training (MRT) in NMT to directly optimize the evaluation metrics. Ayana et al. (2016) apply MRT on abstractive sentence summarization task and the results show that optimizing for ROUGE improves the test performance.

## 7 Conclusion

This paper proposes a selective encoding model which extends the sequence-to-sequence model for abstractive sentence summarization task. The selective mechanism mimics one of the human summarizers' behaviors, selecting important information before writing down the summary. With the proposed selective mechanism, we build an end-to-end neural network summarization model which consists of three phases: encoding, selection, and decoding. Experimental results show that the selective encoding model greatly improves the performance with respect to the state-of-the-art methods on English Gigaword, DUC 2004 and MSR-ATC test sets.

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
