# Peer review of "Selective Encoding for Abstractive Sentence Summarization"

_ACL 2017 — decision unknown_

[Official Review · Reviewer 1 · rating 4 · confidence 3]
soundness 5 · originality 5 · clarity 5 · impact 3 · substance 4 · appropriateness 5 · meaningful comparison 3 · presentation format Oral Presentation

- Strengths:

The authors propose a selective encoding model as extension to the
sequence-to-sequence framework for abstractive sentence summarization. The
paper is very well written and the methods are clearly described. The proposed
methods are evaluated on standard benchmarks and comparison to other
state-of-the-art tools are presented, including significance scores. 

- Weaknesses:

There are some few details on the implementation and on the systems to which
the authors compared their work that need to be better explained. 

- General Discussion:

* Major review:

- I wonder if the summaries obtained using the proposed methods are indeed
abstractive. I understand that the target vocabulary is build out of the words
which appear in the summaries in the training data. But given the example shown
in Figure 4, I have the impression that the summaries are rather extractive.
The authors should choose a better example for Figure 4 and give some
statistics on the number of words in the output sentences which were not
present in the input sentences for all test sets.

- page 2, lines 266-272: I understand the mathematical difference between the
vector hi and s, but I still have the feeling that there is a great overlap
between them. Both "represent the meaning". Are both indeed necessary? Did you
trying using only one of them.

- Which neural network library did the authors use for implementing the system?
There is no details on the implementation.

- page 5, section 44: Which training data was used for each of the systems that
the authors compare to? Diy you train any of them yourselves?

* Minor review:

- page 1, line 44: Although the difference between abstractive and extractive
summarization is described in section 2, this could be moved to the
introduction section. At this point, some users might no be familiar with this
concept.

- page 1, lines 93-96: please provide a reference for this passage: "This
approach achieves huge success in tasks like neural machine translation, where
alignment between all parts of the input and output are required."

- page 2, section 1, last paragraph: The contribution of the work is clear but
I think the authors should emphasize that such a selective encoding model has
never been proposed before (is this true?). Further, the related work section
should be moved to before the methods section.

- Figure 1 vs. Table 1: the authors show two examples for abstractive
summarization but I think that just one of them is enough. Further, one is
called a figure while the other a table.

- Section 3.2, lines 230-234 and 234-235: please provide references for the
following two passages: "In the sequence-to-sequence machine translation (MT)
model, the encoder and decoder are responsible for encoding input sentence
information and decoding the sentence representation to generate an output
sentence"; "Some previous works apply this framework to summarization
generation tasks."

- Figure 2: What is "MLP"? It seems not to be described in the paper.

- page 3, lines 289-290: the sigmoid function and the element-wise
multiplication are not defined for the formulas in section 3.1.

- page 4, first column: many elements of the formulas are not defined: b
(equation 11), W (equation 12, 15, 17) and U (equation 12, 15), V (equation
15).

- page 4, line 326: the readout state rt is not depicted in Figure 2
(workflow).

- Table 2: what does "#(ref)" mean?

- Section 4.3, model parameters and training. Explain how you achieved the
values to the many parameters: word embedding size, GRU hidden states, alpha,
beta 1 and 2, epsilon, beam size.

- Page 5, line 450: remove "the" word in this line? "SGD as our optimizing
algorithms" instead of "SGD as our the optimizing algorithms."

- Page 5, beam search: please include a reference for beam search.

- Figure 4: Is there a typo in the true sentence? "council of europe again
slams french prison conditions" (again or against?)

- typo "supper script" -> "superscript" (4 times)

[Official Review · Reviewer 2 · rating 4 · confidence 4]
soundness 5 · originality 5 · clarity 5 · impact 3 · substance 4 · appropriateness 5 · meaningful comparison 3 · presentation format Oral Presentation

- Strengths:

The paper is very clear and well-written. It proposes a novel approach to
abstractive sentence summarization; basically sentence compression that is not
constrained to having the words in the output be present in the input. 

- Excellent comparison with many baseline systems. 

- Very thorough related work. 

- Weaknesses:

The criticisms are very minor:

- It would be best to report ROUGE F-Score for all three datasets. The reasons
for reporting recall on one are understandable (the summaries are all the same
length), but in that case you could simply report both recall and F-Score. 

- The Related Work should come earlier in the paper. 

- The paper could use some discussion of the context of the work, e.g. how the
summaries / compressions are intended to be used, or why they are needed. 

- General Discussion:

- ROUGE is fine for this paper, but ultimately you would want human evaluations
of these compressions, e.g. on readability and coherence metrics, or an
extrinsic evaluation.

[Official Review · Reviewer 3 · rating 4 · confidence 4]
soundness 5 · originality 5 · clarity 3 · impact 3 · substance 4 · appropriateness 5 · meaningful comparison 3 · presentation format Poster

The paper presents a new neural approach for summarization. They build on a
standard encoder-decoder with attention framework but add a network that gates
every encoded hidden state based on summary vectors from initial encoding
stages. Overall, the method seems to outperform standard seq2seq methods by 1-2
points on three different evaluation sets.

Overall, the technical sections of the paper are reasonably clear. Equation 16
needs more explanation, I could not understand the notation. The specific
contribution,  the selective mechanism, seems novel and could potentially be
used in other contexts. 

The evaluation is extensive and does demonstrate consistent improvement. One
would imagine that adding an additional encoder layer instead of the selective
layer is the most reasonable baseline (given the GRU baseline uses only one
bi-GRU, this adds expressivity), and this seems to be implemented Luong-NMT. My
one concern is LSTM/GRU mismatch. Is the benefit coming from just GRU switch? 

The quality of the writing, especially in the intro/abstract/related work is
quite bad. This paper does not make a large departure from previous work, and
therefore a related work nearby the introduction seems more appropriate. In
related work, one common good approach is highlighting similarities and
differences between your work and previous work, in words before they are
presented in equations. Simply listing works without relating them to your work
is not that useful. Placement of the related work near the intro will allow you
to relieve the intro of significant background detail and instead focus on more
high level.